# Comparative Analysis of rRNA Removal Methods for RNA-Seq Differential Expression in Halophilic Archaea

**DOI:** 10.3390/biom12050682

**Published:** 2022-05-10

**Authors:** Mar Martinez Pastor, Saaz Sakrikar, Deyra N. Rodriguez, Amy K. Schmid

**Affiliations:** 1Biology Department, Duke University, Durham, NC 27708, USA; mar.martinez@duke.edu (M.M.P.); saaz.sakrikar@duke.edu (S.S.); 2University Program in Genetics and Genomics, Duke University, Durham, NC 27708, USA; 3New England Biolabs, Ipswich, MA 01938, USA; rodriguezd@neb.com

**Keywords:** archaea, RNAs-seq, rRNA removal, halophiles, transcriptomics

## Abstract

Despite intense recent research interest in archaea, the scientific community has experienced a bottleneck in the study of genome-scale gene expression experiments by RNA-seq due to the lack of commercial and specifically designed rRNA depletion kits. The high rRNA:mRNA ratio (80–90%: ~10%) in prokaryotes hampers global transcriptomic analysis. Insufficient ribodepletion results in low sequence coverage of mRNA, and therefore, requires a substantially higher number of replicate samples and/or sequencing reads to achieve statistically reliable conclusions regarding the significance of differential gene expression between case and control samples. Here, we show that after the discontinuation of the previous version of RiboZero (Illumina, San Diego, CA, USA) that was useful in partially or completely depleting rRNA from archaea, archaeal transcriptomics studies have experienced a slowdown. To overcome this limitation, here, we analyze the efficiency for four different hybridization-based kits from three different commercial suppliers, each with two sets of sequence-specific probes to remove rRNA from four different species of halophilic archaea. We conclude that the key for transcriptomic success with the currently available tools is the probe-specificity for the rRNA sequence hybridization. With this paper, we provide insights into the archaeal community for selecting certain reagents and strategies over others depending on the archaeal species of interest. These methods yield improved RNA-seq sensitivity and enhanced detection of low abundance transcripts.

## 1. Introduction

The expression of the genomic information of an organism depends on the cell status and environmental factors that determine the phenotype. The genes across the genome that are being transcribed collectively define the transcriptome, and the compendium of methods that enable the study of the expression of large number of genes simultaneously is known as transcriptomics. RNA sequencing (RNA-seq) has emerged as a widely used approach to transcriptome profiling with high throughput, sensitivity, dynamic range, and relatively low cost compared to former methods such as microarrays. The first successful RNA-seq experiments were performed using eukaryotic model organisms [1,2,3,4]; however, using this tool for understudied models such as archaea has been challenging despite their biological and evolutionary importance.

Archaea are prokaryotic microorganisms that were defined as the third branch of life in the late 1970s, when Carl Woese and colleagues found substantial 16S differences that warranted classifying the Archaea as a distinct group separate from the Bacteria and Eukarya [5]. Recent phylogenetic evidence is more consistent with a two-domain tree, with Eukarya stemming from Archaea [6]. Although archaeal species are typically known for their survival in extreme environments, they are now known to be diverse and abundant, colonizing a vast array of habitats (from oceans to human skin to extreme environments [7,8]). Therefore, differential expression analysis using transcriptomics in archaea is an important step for a better understanding of responses to diverse environments [9,10]. Such studies advance knowledge of the unique molecular biology of archaea, which combine the molecular characteristics of both bacteria and eukaryotes, such as transcriptional regulation [11].

Despite previous progress on differential expression by RNA-seq in archaea [12], this method has recently become unavailable. Previously, archaeal transcriptomics studies successfully depleted rRNA using commercially available reagent kits for rRNA removal in bacteria [13,14,15,16,17]. However, these kits were discontinued in 2018. Ribosomal RNA (rRNA) in archaeal transcriptomes can reach more than 90% of the total cellular RNA. As we report in the current work, ribodepletion is a key step for reliable RNA-seq results because high rRNA sequencing reads can preclude the detection of messenger RNA (mRNA). rRNA removal enables higher sequencing depth of mRNA, leading to better detection of transcripts. This is critical for analyzing differential expression, particularly when detecting non-coding or lowly expressed RNAs [18]. Previous studies have suggested a minimum sequencing depth of two [19] to ten [20] million reads per sample for obtaining reproducible results for differential expression, while the ENCODE consortium [21] mandates 30 million reads (albeit for much larger human genomes). Such sequencing depth enables sound statistical comparisons of differential expression on a per-gene basis: at least five reads per gene are typically needed to detect the significance of change in expression for a given gene [18]. Some archaeal studies have reported RNA-seq without rRNA removal, but these were conducted for different purposes that are possible without rRNA removal (e.g., transcription start site mapping [22], small RNA detection [12], etc.). Removing rRNA also substantially reduces the cost of RNA-seq, enabling extensive sample multiplexing in a single sequencing run, especially for relatively small archaeal genomes (~2–8 Mbp).

In this work, we have studied four species of halophilic archaea that have been widely used as model organisms in the archaeal research community: *Halobacteium salinarum* (HBT) and *Haloarcula hispanica* (HAH) of the family Halobacteriaceae require salt concentrations close to saturation, whereas *Haloferax volcanii* (HVO) and *Haloferax mediterranei* (HFX) of the family Haloferacales colonize lower salinity environments. These four species are highly tractable models for extremophilic microorganisms given their relatively fast generation time (2–6 h in rich medium), facile genetic tools [23,24,25], and highly curated genomic annotations and databases [26,27,28,29]. Establishing a set of tools and best practices for transcriptomics methods would, therefore, greatly facilitate advances in this field.

Archaeal RNA, similar to that of bacteria, lacks a 3′ polyA tail, so rRNA cannot be removed by polyT tagging. Here, we test two methodologies for rRNA depletion in archaea using (a) biotinylated probes and (b) enzymatic digestion. We use probes that come packaged with commercial kits as well as sequence-specific probes customized for particular species of interest. The first approach (biotinylated probes/streptavidin beads) consists of a physical removal of rRNA by hybridizing with a pool of biotinylated oligo probes. These probes are then captured and removed from the RNA sample using streptavidin-coated magnetic beads. In contrast, the enzymatic removal of rRNA consists of generating DNA-rRNA hybrids by incubating specifically designed DNA probes complementary to rRNA. Hybrids are then treated with RNaseH that catalyzes the cleavage of RNA when it is bound to a DNA substrate.

Here, we report that the two methods are equally successful for removing rRNA across the four species of halophilic archaea growing in diverse media. Both methods can be used successfully with probe sequences custom-designed for one species or with a broad probe pool designed to target multiple species simultaneously. We show that bacterial rRNA probes are sufficiently divergent in sequence to preclude the use of recently developed custom and commercial bacterial rRNA probe sets in archaea [30]. These methods are robust to varying culturing conditions (rich and defined media). This analysis has achieved the goal of identifying an efficient and broadly useful strategy for depleting undesirable archaeal rRNA prior to sequencing for successful transcriptomics.

## 2. Material and Methods

### 2.1. Media, Strains, and Growth Conditions

All strains, media, and conditions used for this study are summarized in Table 1 and Table 2.

For routine culturing in these media, each species was freshly streaked onto solid medium from frozen stock. Single colonies were inoculated in triplicate in 3 mL of rich, minimal, or defined liquid media (Table 2) and grown aerobically until saturation (stationary phase) at 42 °C with continuous shaking at 225 rpm for HBT, HFX, and HVO; 37 °C for HAH. From each saturated pre-culture, 50 mL cultures were initiated by diluting the pre-culture to OD_660_ = 0.02–0.1 in 150 mL Pyrex flasks, and 3 mL of each were harvested in mid exponential phase OD_660_ = 0.4–0.8 (doubling times and incubation times included in Table 3), by centrifugation in a tabletop centrifuge (5424, Eppendorf) at 21,130× *g* for 3 min. Supernatant was discarded and pellets were immediately snap-frozen in liquid N_2_ and stored no longer than 3 weeks at −80 °C until RNA extraction.

### 2.2. RNA-Seq Experimental Protocol

Total RNA was extracted from pellets using Absolutely RNA Miniprep kit (Agilent Technologies, Santa Clara, CA, USA) according to manufacturer’s instructions. The concentration and integrity of resultant RNA was quantified by Nanodrop One (Thermo Scientific, Grand Island, NY, USA) and RNA electropherograms, Bioanalyzer 2100 Instrument with the RNA 6000 Nano kit (Agilent Technologies, Santa Clara, CA, USA), respectively. RNA was checked for DNA contamination by PCR using 200–300 ng of input RNA and primers given in Table 4 for 30–35 cycles. Extracted RNA was high quality in all samples, with a Bioanalyzer RNA integrity number (RIN) greater than 8.

Ribosomal RNA was removed using the following reagent kits and methodologies, abbreviated throughout the text and figures as indicated below:Biotinylated probes with strepdavidin bead pull-down:
Discontinued Ribo-Zero rRNA Removal Kit (Bacteria). Abbr: RZ;siTools HVO RiboPOOL^TM^ with probes specific for HVO. Abbr: rP-HVO;siTools Pan-Archaea riboPOOL^TM^ (probes included). Abbr: rP-PA.RNAse H and enzymatic depletion-based protocols with magnetic bead pull-down:
Ribo-Zero Plus Kit (probes included). Abbr: RZ+;NEBNext Bacteria rRNA depletion Kit (New England Biolabs) with probes designed for bacteria (included in kit from NEB). Abbr: NEB-B;NEBNext Depletion Core Reagent Set with customized sequence-specific probes for HVO (Appendix A). These probes were designed using the NEB web tool (https://depletion-design.neb.com/, accessed on 3 January 2020) and ordered from IDT technologies (idtdna.com, accessed on 3 January 2020). Abbr: NEB-HVO.



RNA input to each depletion kit was 300–500 ng. Ribodepletion was performed according to the manufacturer’s manuals using default or custom-designed probes as well as modifying time of enzymatic incubation with RNaseH. These details and ordering information are specified in Appendix A.

Library preparation from 1–10 ng rRNA-depleted RNA was performed using NEBNext UltraII Directional RNA Library Preparation Kit (Illumina, #E7760) following the vendor protocols and complementing cleaning steps with NEBNext Sample Purification Beads (#E7767). An extra cleaning step using the same type of beads was carried out when samples showed contamination with adaptor dimers. The obtained library quality and concentration was assessed by monitoring the distribution of the fragment sizes with a Bioanalyzer 2100 instrument using RNAnano reagent kit (Agilent Technologies, Santa Clara, CA, USA). This size and quantity information was used for pooling the libraries in equimolar concentrations to normalize each library. Libraries were subjected to HiSeq2500, HiSeq4000, or NovaSeq6000 by the Sequence and Genomics Technologies Facility at Duke University. Additional experimental metadata, results, and details are given in Appendix A.

### 2.3. Data Analysis

#### 2.3.1. Publications on Archaeal RNA-Seq per Year

Data regarding the number of yearly publications, available from the National Center for Biotechnology Information (NCBI) PubMed database (https://pubmed.ncbi.nlm.nih.gov/, access date 4 May 2021), were searched for with the phrases “archaea”, “RNA-Seq”, and “archaea RNA-Seq”. Database hits were downloaded from the NCBI PubMed database on 1 November 2021. The publication of Carl Woese’s seminal paper regarding the classification of Archaea in 1977 [5] was used as the starting date. The downloaded data are shown in Appendix A. The code used to generate Figure 1 is in https://github.com/amyschmid/rRNA_analysis.

#### 2.3.2. RNA-Seq Data Processing

FASTQ files generated by sequencing were downloaded and processed as described previously [34]. Files were quality-checked using FastQC, adapter sequences were trimmed using TrimGalore! with cutadapt (FastQC and TrimGalore! downloaded from http://www.bioinformatics.babraham.ac.uk/projects/, accessed on 13 November 2017). Trimmed files were aligned to the reference genomes of the four species of interest (Table 1) using Bowtie2 [35]. The resultant SAM files were converted into a compact BAM file using SAMtools [36] to generate, sort, and index reads. BAM files were used as the input for HTSeq-count [37] to generate a count file, assigning a numeric raw count of reads to each gene. Details regarding the full workflow are included in reference [34]. To determine the rRNA percentage remaining following depletion, the counts corresponding to each of 16S, 23S, and 5S rRNA genes was divided by the total number of raw counts mapping to all genes. The ratio was multiplied by 100 to yield a percentage. These genes are listed in Table 5.

The results, expressing all rRNA, 16S rRNA, 23S rRNA, and 5S rRNA as a percentage of total reads, are listed in Appendix A. The code used to generate figures is in https://github.com/amyschmid/rRNA_analysis, and the input to the code is also given in Appendix A under the appropriate tabs.

#### 2.3.3. Probe Specificity Analysis

Sequences of probes custom-designed for HVO rRNA removal using the NEB website (https://depletion-design.neb.com/) were compared to HBT strain NRC-1 genome sequence using NCBI BlastN search with default parameters (NCBI taxonomy ID: 64091; NCBI access date 4 May 2021). The resultant sequence identity (expressed as a percentage) was noted for each of the 117 sequences. These data were classified into four categories: 100% identity, 90–99% identity, <80% identity, and no significant similarity. The probe sequences, BLAST results, and identity percentages are listed in Appendix A. The code used to generate the corresponding figure is in https://github.com/amyschmid/rRNA_analysis and the specific inputs to generate this figure are in the appropriate tabs within Appendix A.

#### 2.3.4. Count Correlations

RNA-seq read counts corresponding to all genes outside of rRNA genes for different rRNA removal methods and replicates in HBT and HVO were calculated as described above. Each gene’s count was expressed as a percentage of total counts, and the arithmetic average of all replicates using a particular method was calculated. These average values for each gene for a given removal method were then noted in Appendix A. The code used to generate the corresponding figure is in https://github.com/amyschmid/rRNA_analysis and the specific inputs to generate this figure are in the appropriate tabs within Appendix A.

#### 2.3.5. Power Analysis

RNA-Seq data generated from a pilot run for a published project [34] from the Schmid lab were used as input to the power optimization tool Scotty [38]. Instructions for accessing the Scotty web app can be found here: https://github.com/amyschmid/rRNA_analysis/scotty-access-instructions. This tool was used to assess power for differential expression experiments involving up to 10 biological replicates with between 1 and 15 million reads mapping to genes for each replicate, so that at least 75% of two-fold differentially expressed genes could be detected at *p* < 0.01.

#### 2.3.6. Probe Design for Other Species of Interest

To facilitate future studies, probes were also designed for the remaining three species of interest (HBT, HAH, and HFX), and FASTA files for gene sequences encoding 16S and 23S rRNA were downloaded from NCBI Gene database (https://www.ncbi.nlm.nih.gov/gene; accessed on 28 April 2022) and submitted to the NEB online design tool (https://depletion-design.neb.com/). Probe set sequences and gene accession numbers are given in Appendix A.

## 3. Results

### 3.1. Discontinuation of the Illumina RiboZero Kit Is Associated with a Decline in Published Archaeal RNA-Seq Studies

RNA-Seq of archaeal species belonging to diverse clades has previously been facilitated by rRNA depletion using the bacterial Ribo-Zero kit from Illumina [13,14,15,16,17] (Methods). However, the kit was discontinued in 2018. To determine the impact of this discontinuation, we conducted a comprehensive literature search on the PubMed database for articles reporting on archaea (1977–present) and on RNA-seq in archaea (2010–present). The discontinuation of the Ribo-Zero kit appears to correlate with a plateau and decline of papers published on the topic of RNA-seq in archaea, even as the number of publications on archaea in general and on RNA-Seq in other domains of life has grown (Figure 1). Within our lab, we had successfully used this kit on two model halophile species, HBT [14] and HVO (Mar Martinez Pastor, unpublished data). The Ribo-Zero kit used biotinylated RNA probes designed to deplete abundant rRNA transcripts from bacterial total RNA with streptavidin beads. We observed 100% removal of rRNA from HVO total RNA samples (Figure 2). In contrast, removal from HBT was variable, with a median rRNA value of 35% (range 18.7–46.4%; Figure 2; Appendix A), at a level which allowed analysis of differential expression [14]. Because RNA-seq transcriptomic profiling studies across halophilic archaea are valuable to understand responses to environmental perturbation, we were motivated to find a suitable replacement capable of matching or bettering this performance across four model species of halophiles routinely used in our lab (HBT, HVO, HFX, and HAH, abbreviations listed in Table 1).

### 3.2. Testing New rRNA Depletion Strategies on Total RNA Samples from Halobacterium salinarum (HBT)

In our lab, discontinuation of the Ribo-Zero kit stalled ongoing RNA-seq experiments across four species of halophilic archaea (HBT, HVO, HAH, and HFX). We started with rRNA removal in HBT given our recent success with RNA-seq in this organism [34]. We used three enzymatic digestion-based rRNA depletion approaches from the following commercial kits (details in Appendix A and Methods): (a) NEBNext Bacterial rRNA Removal Kit (probes included, abbreviated throughout as “NEB-B”); (b) NEBNext rRNA Core Depletion Reagent Set (with user-designed probes specific for HVO, a method abbreviated throughout as “NEB-HVO”); and (c) the newly released Ribo-Zero Plus kit from Illumina (includes probes allowing universal depletion across bacteria and eukaryotes, “RZ+”). We used HVO-specific probes at first in HBT to determine the effectiveness of using species-specific probes to remove rRNA across related halophilic archaea. Following rRNA removal, the resultant RNA samples were subjected to Next Generation sequencing, and the number of rRNA reads removed was quantified as compared to an untreated RNA control (Methods).

We observed that ~95% of reads from sequenced untreated RNA correspond to rRNA (Figure 3, Appendix A). RZ+ treatment achieved a negligible reduction of rRNA to ~92%. A slightly more substantial reduction was seen with the NEB-B method, with a median remaining rRNA percentage of 86%. Of these methods, the best results were obtained using NEBNext with customized probes designed to bind HVO rRNA sequences (NEB-HVO), although high levels of rRNA still remained (median remaining rRNA 80.5%, range 63% to 86%). We note that using no removal, RZ+, and NEB-HVO methods result in a range of ~1.5–3.6M reads mapping to non-rRNA genes per sample (with 12 total samples run on one lane, Appendix A). Based on our power analysis using online tools [38], this level of sequencing depth would require 5–6 biological replicates for reliable detection of 75% of differentially expressed genes (FDR < 0.05, log fold change ≥ 2.0) (Appendix A). Since this depth was achieved with 12 samples multiplexed per lane, a requirement of 4–5 samples of each type would restrict RNA-Seq experimental design to a single comparison (for example, two genotypes in one condition or two conditions for the same genotype) per lane. Hence, the inefficient rRNA removal severely limits the extent to which samples can be multiplexed, increasing costs even in the modern high-throughput sequencing instruments used here (Appendix A).

We hypothesized that poor rRNA removal may stem from either the incomplete RNase H digestion or the imperfect sequence match between the HVO rRNA probes used in the NEB-HVO method and the rRNA genes of HBT. To test the efficacy of RNAse H digestion, we carried out this digestion over 30 min (manufacturer protocol) and 120 min (extended digestion) using the NEB-HVO method. Each digestion time used the same extracted RNA sample (split into two different aliquots for digestion) and was performed in biological triplicate within the same sequencing batch. A marked improvement in rRNA removal is seen in the 120-min digestion (Figure 4A), with 75% median rRNA remaining, as compared to 85% for the 30 min samples. However, when comparing the results between different batches of sequencing, we found that the batch effect was stronger than the RNAse effect: 30 min RNAse H digestion from a different batch produced a rRNA range of 63–76% (median 68%), better than even the 120 min digestion from the first batch. Hence, while longer RNAse H digestion could potentially improve rRNA removal, this effect is inconsistent.

Based on these results, we then tested the hypothesis that this relatively poor rRNA removal (compared to the discontinued RZ method) was associated with sequence mismatches between probes and rRNA. Using the NEB-HVO method, we observed that the probe sequences custom-designed for HVO rRNA matched HBT 16S rRNA sequences better than 23S probe sequences (Figure 4B, Appendix A). We note that using the NEB custom probe method allows for the study of probe sequences, unlike the other methods, where probe sequences are not provided by the company. In total, 19% of 16S HVO probe sequences had 100% identity with HBT 16S rRNA, compared to only 8% for 23S rRNA. Conversely, 25% of 23S probe sequences shared no sequence similarity with HBT 23S rRNA, while this was only 16% for 16S. Corresponding with these different levels of sequence identity, we observed that 16S rRNA removal was more effective (~15–35% remaining) than 23S rRNA removal (~55–65% remaining, Figure 4C, Appendix A). Bacterial rRNA sequences (from which probe sequences included in the NEB rRNA removal kit are derived) are very different from those of archaea. Thus, as expected, the percentage of 16S and 23S rRNA reads remaining following treatment with the NEB-B method was comparable to the no-removal control (Figure 4C). Hence, there is a strong relation between probe sequence and rRNA removal, with even slight increases in probe specificity (Figure 4B), resulting in profound differences in rRNA removal (Figure 4C).

We conclude from these experiments that the NEBNext Core Reagent Set kit with probes custom-designed for the related species HVO (NEB-HVO) is the best of the reagent kits that we tested for HBT rRNA removal. RiboZero Plus (RZ+) and NEBNext Bacterial kit using the bacterial probes (NEB-B) led to less efficient rRNA removal for HBT. We expect that targeting custom probes specifically designed for the species of interest would likely result in better rRNA removal. To facilitate advancement in rRNA removal methods for the four halophilic archaeal species of interest here, we have used the NEB online design tool to develop custom sets of probe sequences for each of HBT, HVO, HAH, and HFX (Methods, Appendix A).

### 3.3. Species-Specific Probe Methods Efficiently Remove Haloferax volcanii (HVO) rRNA

Having shown the importance of probe sequence specificity, we next tested two different methods with rRNA probes targeted to HVO against HVO total RNA samples: (a) NEBNext Core Reagent Set (“NEB-HVO” method); and (b) the siTools RiboPool kit (“rP-HVO”). Unlike the enzymatic NEB-HVO method, rP-HVO uses streptavidin-based removal of rRNA hybridized to biotinylated probes. For both methods, we used probes custom designed to be specific to HVO rRNA sequences (see Methods). We observed that both methods achieved nearly complete rRNA removal: median values of 0.008% and 0.000008% rRNA remaining were observed using NEB-HVO and rP-HVO methods, respectively (Figure 5; Appendix A). These results with near-complete rRNA depletion in HVO with species-specific probes is in line with the observations above: the limiting factor with these probe-based methods is the identity of probe sequences with target rRNA sequences. Overall, we found that using probes targeted to HVO with either method resulted in efficient and near-complete removal of rRNA from HVO samples. 

### 3.4. siTools Panarchaea Kit Efficiently Removes rRNA from Diverse Halophilic Archaeal Species

To expand our analysis to other model species, we then tested the siTools riboPOOL Panarchaea kit (rP-PA, Appendix A, methods). The probe set associated with this kit is composed of high complexity pools of biotinylated DNA probes with sequences designed to deplete rRNA from a broad spectrum of archaea, including several classes of Euryarchaeota and Proteoarchaeota (https://sitoolsbiotech.com/ribopools.php). The Panarchaea riboPOOL probes have been shown to remove 99% of rRNA from *Sulfolobus solfataricus* and *Sulfolobus acidocaldarius* (https://sitoolsbiotech.com/pdf/microbes-ribopools-072021.pdf), but to our knowledge, they have not been published for euryarchaeal species such as the four model halophiles of interest here. After using this kit for ribodepletion, we observed that all tested RNA samples across the four species contained <10% rRNA, with median values of 3.3%, 0.0002%, 0.04%, and 0.5% for HBT, HVO, HFX, and HAH, respectively (Figure 6, Appendix A). This extensive rRNA removal is more effective for HBT than for any previously tested methods (Figure 2 and Figure 3), and equally as effective as NEB-HVO and rP-HVO methods for HVO (Figure 4 and Figure 5). The other two species had not been previously tested, and no other RNA-seq results are available for comparison in the literature (other than for HFX small RNAs: in that study rRNA was not removed [39]). Taken together, these results demonstrate that the Panarchaea method (rP-PA) efficiently removed rRNA for four different model species of halophilic archaea.

### 3.5. Choice of Removal Method Does Not Affect Per-Gene Read Counts

It was observed previously that using different rRNA removal methods can affect the relative read counts of some non-rRNA genes [40,41]. Therefore, we tested whether rRNA removal and the choice of removal method changes the relative levels of mRNA. We calculated the gene counts from each sample as a percentage of the total (non-rRNA) counts from that sample, and correlated these relative counts obtained from different rRNA removal techniques (see Methods, Figure 7). We observed strong correlations of normalized relative counts of non-rRNA genes among different rRNA removal methods, as well as with untreated total RNA data for HBT (Figure 7A). The Pearson’s correlation coefficients between per-gene normalized read counts across different methods used on HBT were in the range 0.91–0.99, with an average value of 0.95. Correlation with control (untreated) samples was >0.92. Similar results were seen with HVO: 0.94–0.99, average 0.97 (Figure 7B, Appendix A). Note that, in HVO, the correlation of all methods to RZ is slightly lower (>0.940, Figure 7B). We surmise that this is due to differences in growth medium—RZ samples were prepared in minimal medium, whereas all other samples were prepared in rich medium (Table 2). Nonetheless, this correlation is still high and significant. Based on this analysis, we conclude that rRNA removal and the choice of removal method does not change the number of reads on a per-gene basis in halophilic archaea. These rRNA removal methods can, therefore, be used for downstream applications, such as differential gene expression analysis.

### 3.6. Utility of rRNA Removal Is Seen in Counts of Non-rRNA Genes

Previous studies have suggested a minimum sequencing depth of two [19] to ten [20] million reads per sample for obtaining reproducible results for differential expression. On a per-gene basis, five reads is considered a threshold, below which differential expression analysis is unreliable [18]. We sought to understand how rRNA removal affects transcript detection using data from HBT, from which we have data for a wide array of rRNA removal methods (including no removal), and a large range of rRNA remaining in sequenced samples (2–95%). Across these samples, we calculated the number of annotated genes with no mapped reads as well as <5 mapped reads. For consistency, we only considered samples that had been sequenced on the same machine (NovaSeq6000). We observed that more complete rRNA removal generally leads to increased detection of genes (Figure 8). All numbers that follow are median values, obtained from Appendix A. For untreated RNA (~95% rRNA), 320 genes showed no reads, and this reduced to 312 for RNA treated with NEB-B (~86% rRNA) and further to 307 with NEB-HVO (~68% rRNA). The Panarchaea kit (rP-PA) reduced the number of undetected genes to 298 (~3% rRNA). For genes with <5 counts detected, there was a more dramatic change, from 385 genes for untreated RNA, but only 310 for Panarchaea kit. This trend held true even though the total number of reads for all genes (including rRNA) had relatively similar median values of ~30 M and ~27 M reads, respectively (Appendix A), suggesting that the improved detection of lowly expressed genes is associated with more complete rRNA removal rather than deeper overall sequencing of the samples. These above results indicate that better rRNA removal improves per-gene counts and detection of lowly expressed genes, which is important for making accurate statistical conclusions regarding differential expression. rRNA removal enables increased multiplexing, and therefore, reduced cost for RNA-seq experiments.

## 4. Discussion and Conclusions

The main technical challenge for prokaryotic transcriptomics is the low mRNA:rRNA ratio. Historically, different methods have been used to eliminate rRNA without biasing mRNA reads: digestion with exonucleases that preferentially degrade rRNA relying on 5′ monophosphate; subtractive hybridization that captures rRNA binding to antisense oligonucleotides [42,43]; and poly(A) tail addition to discriminate rRNA or reverse transcription with rRNA primers followed by RNaseH digestion [30,44]. However, to our knowledge, none of these methods has been successfully utilized for haloarchaea. Until the end of 2018, the Ribo-Zero kit from Illumina, based on sequence-specific biotinylated probes that hybridize with a pool of microbial rRNA sequences and then selectively remove the hybrids using streptavidin-coated magnetic beads, enabled the removal of ~70% of rRNA for several other archaeal species [13,14,15,16,17]. After this commercially available kit was discontinued, archaeal transcriptomics was at an impasse. Here, we invested time to troubleshoot this problem, where we test and directly compare newly available tools to help the archaeal community to move on with transcriptomic studies. Our investigation provides a guide for choosing a suitable application depending on the model organism or the combination of archaeal species of interest (e.g., communities, labs using multiple cultured species, metatranscriptomics).

We found that both RNAseH-based and biotin-based methods are efficient for rRNA removal. Certain commercially available kits from NEB and siTOOLs are the most effective when probes are designed that target archaeal species of interest. For HVO, the RiboPool kit as well as the NEBNext kit with custom-designed probes that target HVO (Figure 5) resulted in nearly complete rRNA removal. A similar number of total reads was observed after sequencing with no detectable bias in lowly expressed transcripts (Figure 8). In general, when using targeted kits, we found that the most important factor in determining rRNA removal efficiency was percentage identity of the target rRNA with the probe sequence (Figure 4 and Figure 5). We found that the Panarchaea kit from siTOOLs provides very good rRNA depletion across all four species tested here (Figure 6) and we anticipate that these can be effectively used for metatranscriptomics of archaeal communities. Targeted methods from both NEB and SiTOOLs as well as the Panarchaea probe set provide comparable performance to the discontinued RiboZero kit for HVO, with remaining rRNA close to 0. We further note that the Panarchaea kit exceeds the performance of the previous RZ method for HBT (Appendix A, Figure 2 vs. Figure 6). In the future, continuing to deposit raw RNAseq data from the archaeal community into online data repositories such as NCBI Gene Expression Omnibus is critical for progress in the area of transcriptomics, which would facilitate future efforts to predict rRNA removal success depending on probe sequence identity.

One of the most important advantages of choosing an efficient rRNA removal method is analyzing differential expression of low-count genes. In the current study, we show that efficient rRNA depletion enables increased detection of lowly expressed genes (Figure 8). This improvement in coverage of low-count genes enables correct statistical analysis of differential expression [19,20,21]. Accurate detection of lowly expressed transcripts is also important when using RNA-Seq to map the transcriptome [12,45], including in metatranscriptomic protocols [44].

The rapid pace of discovery of new archaeal species [6,46] as well as the use of novel archaeal model organisms in the lab will bring further challenges for transcriptomics experiments. However, the methods tested here provide sufficient flexibility to solve such challenges. For example, it is possible that newly identified archaeal species may encode rRNA sequences divergent from commercially available primer sets such as siTOOLs Panarchaea. The NEBNext Core Reagent Set using the custom probe design tool (https://depletion-design.neb.com/) would therefore be an appropriate choice in this case. To facilitate rRNA removal in halophilic archaea, we have used this tool to design probe sets specific for each of the four species of interest here (HBT, HVO, HAH, and HFX, Appendix A). Probe design for other archaeal species is readily carried out as described in Methods Section 2.3.6. Removal of rRNA enables increased detection of rare transcripts and extensive multiplexing. The methods tested here will, therefore, facilitate rapid progress in understanding the transcriptional response of a wide diversity of archaea to their environment.

## Figures and Tables

**Figure 1 biomolecules-12-00682-f001:**
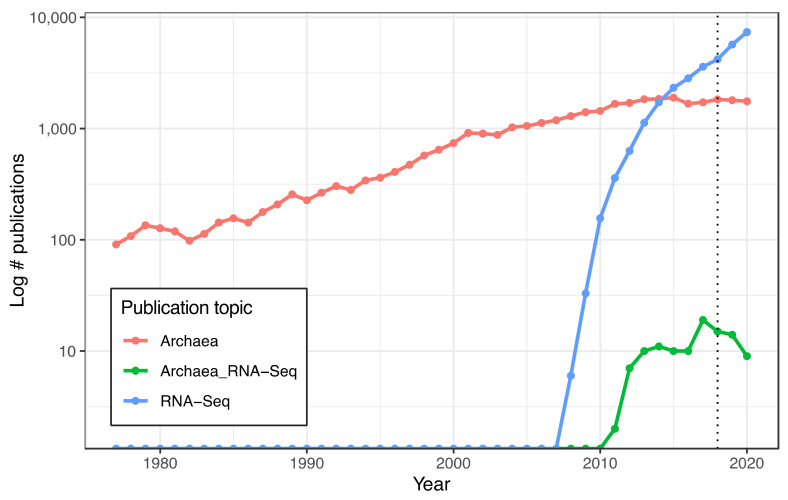
**Slowdown in Archaeal RNA-Seq publications in recent years**. Lines depicting number of publications per year detected in the NCBI PubMed databased searched with the terms “Archaea” (red), “RNA-Seq” (blue), and “Archaea RNA-Seq” (green), plotted on log-scale y-axis. Dotted line at 2018 marks discontinuation of the Illumina RiboZero kit.

**Figure 2 biomolecules-12-00682-f002:**
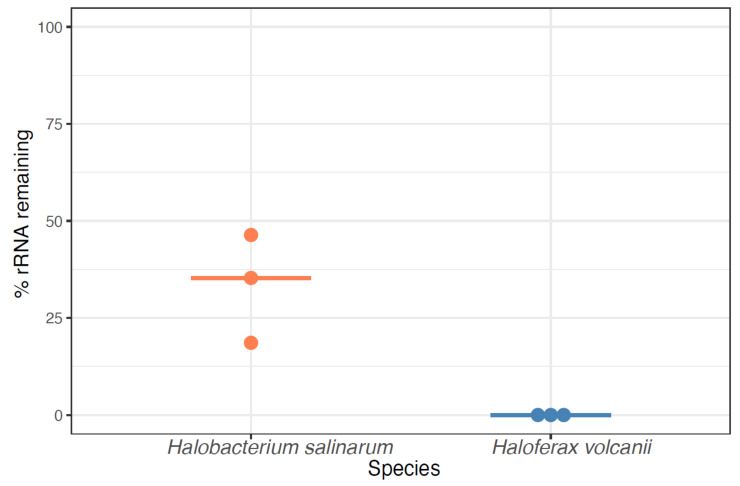
**Percentage of rRNA remaining in halophile RNA by using the discontinued Ribozero kit (RZ).** Each dot denotes one sample, with light orange dots representing *Hbt. salinarum* and blue dots representing *Hfx. volcanii* samples. Horizontal bars represent the median percentage of rRNA remaining.

**Figure 3 biomolecules-12-00682-f003:**
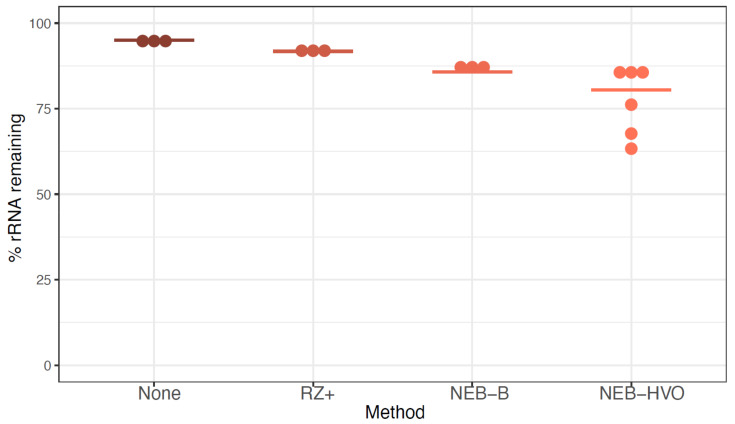
**rRNA removal using alternative methods in *Hbt salinarum*.** Each dot represents the percentage of counts mapping to rRNA genes after using no removal (brown), New Ribozero kit (RZ+, dark orange), NEBNext kit with bacterial probes (NEB-B, orange), and NEBNext kit with HVO probes (NEB-HVO, peach). Horizontal bars represent the median value.

**Figure 4 biomolecules-12-00682-f004:**
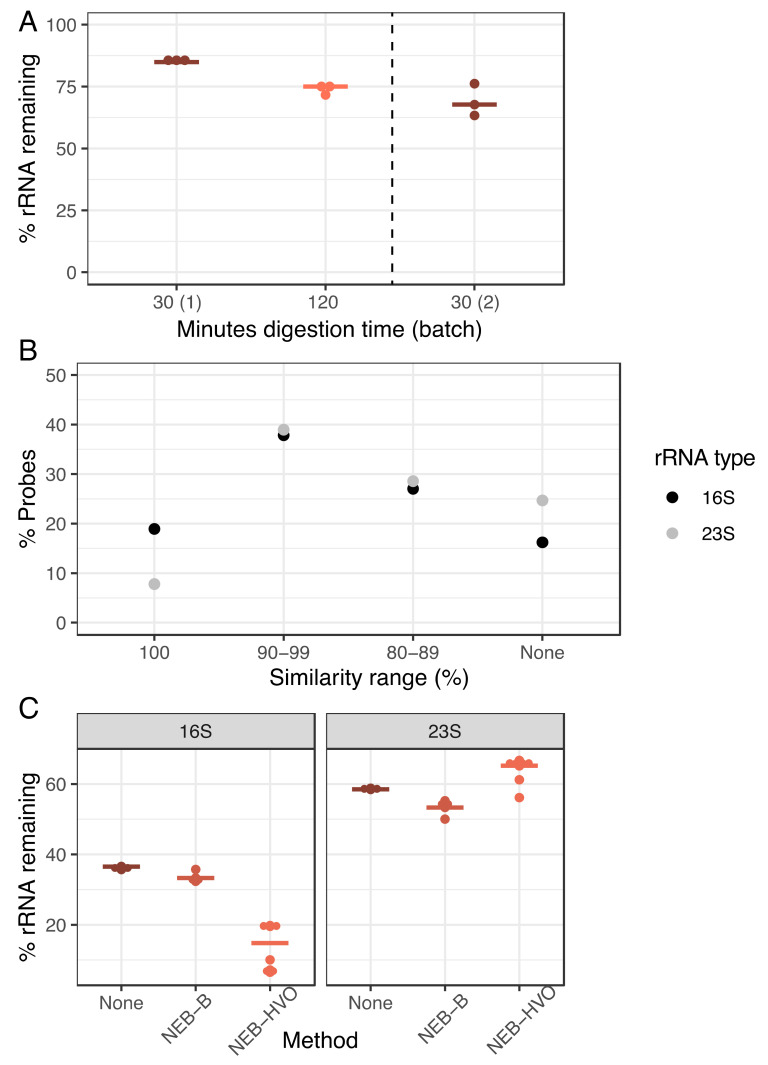
**Increasing RNAse digestion time is less important than probe sequence identity for efficient rRNA removal.** (**A**) Dotplot showing percentage of counts mapping to rRNA genes after using the NEB-HVO method on HBT total RNA samples after 30 min (brown) or minutes 120 (light orange) of RNAseH digestion. “NEB30 (2)” samples to the right of the dotted line were processed and sequenced in a different batch. Horizontal bars represent the median value. (**B**) Percentage of custom-designed HVO probes classified into 16S (black) and 23S (grey). Levels of sequence identity of HVO probes with *Hbt. salinarum* (HBT) 16S and 23S rRNA genes are shown on the X-axis, whereas the percentage of total probes at each sequence identity level is shown on the Y-axis. (**C**) Percentage of total reads mapping to either 16S (left panel) or 23S rRNA (right panel) genes of HBT using three different rRNA removal method—none (brown), NEB-B (dark orange), and NEB-HVO (light orange).

**Figure 5 biomolecules-12-00682-f005:**
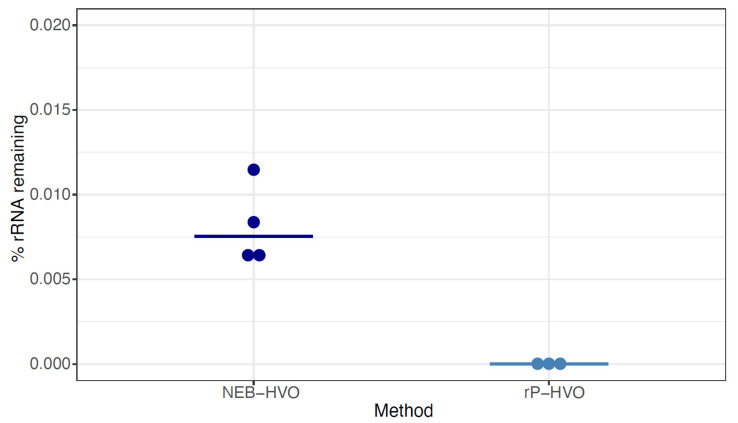
**Species-specific probes efficiently remove rRNA from target species.** Dotplots showing the percentage of rRNA remaining after using probes with sequences specific for *Hfx. volcanii* (HVO) rRNA. Dark blue dots represent %rRNA remaining in individual replicate samples depleted with NEBNext Core Reagent Set (“NEB-HVO” method). Light blue dots represent %rRNA remaining in individual replicate samples depleted with the siTools RiboPool kit (“rP-HVO”). Horizontal bars represent the median value.

**Figure 6 biomolecules-12-00682-f006:**
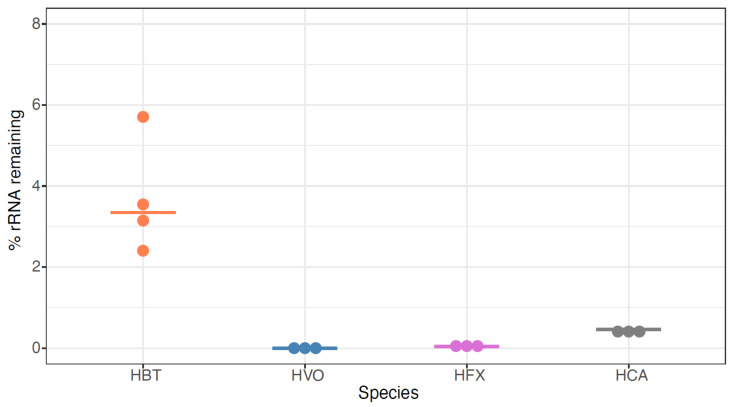
**Panarchaea (rP-PA) kit efficiently removes rRNA from total RNA across halophilic species.** Dotplots showing the percentage of the remaining counts mapping to rRNA genes in *Hbt. salinarum* (HBT, orange), *Hfx. volcanii* (HVO, blue), *Hfx. mediterranei* (HVO, purple), and *Hca. hispanica* (HCA, grey). Horizontal bars represent the median value of three biological replicate samples.

**Figure 7 biomolecules-12-00682-f007:**
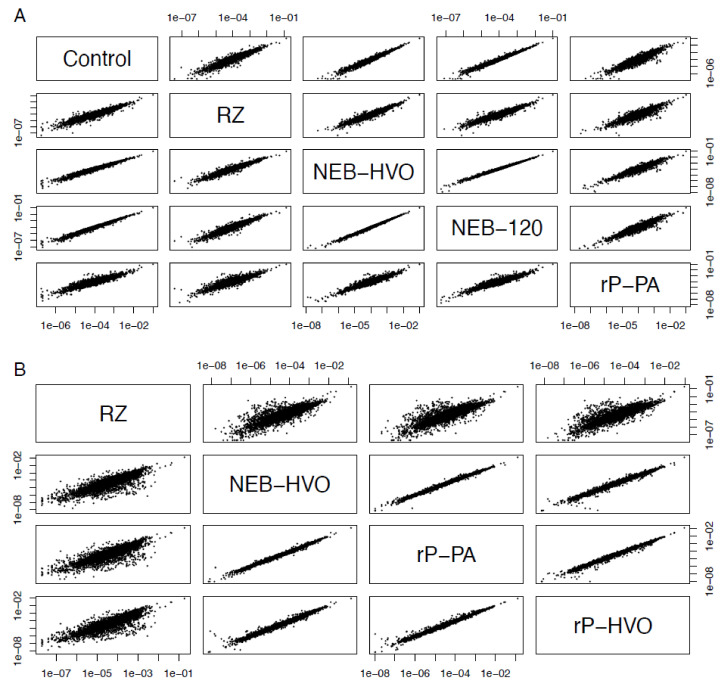
**Choice of removal method does not affect relative abundance of mRNAs.** Correlations between relative abundance of each gene after different rRNA removal methods in (**A**) *Hbt. salinarum* (HBT) and (**B**) *Hfx. volcanii* (HVO). Each dot represents the percent of total normalized reads for each gene (see Methods section). Methods shown here are “Control” (no removal), “RZ” (using discontinued RiboZero kit), “NEB-HVO” (using NEBNext kit with custom HVO probes), “NEB-120” (NEBNext kit with custom HVO probes and 120 min of RNAse digestion), “rP-PA” (siTools riboPOOL method using Panarchaeal probes), and “rP-HVO” (siTools riboPOOL method using HVO-specific probes).

**Figure 8 biomolecules-12-00682-f008:**
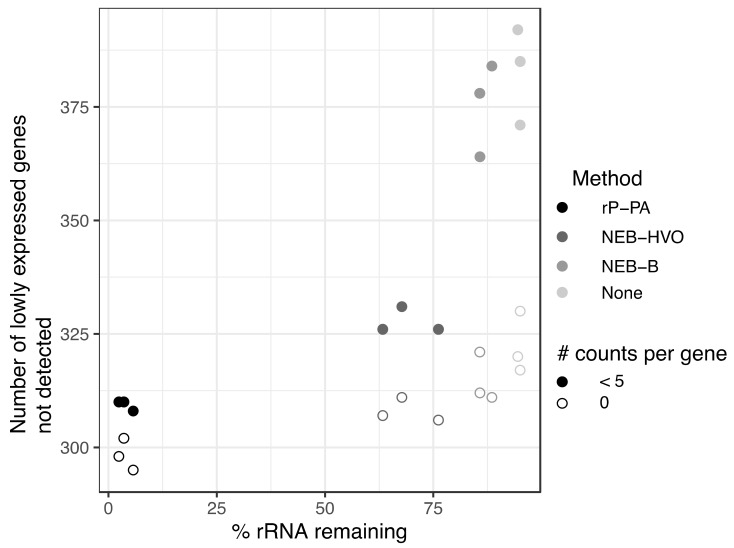
**More complete rRNA removal leads to increased detection of lowly expressed genes.** Number of genes with zero (open circles) and <5 (filled circles) reads detected in sequencing samples treated with different rRNA removal methods in *Hbt. salinarum*. The darker the circle color, the more complete the rRNA removal for each method: riboPOOL Panarchaea (rP-PA, black); NEBNExt with HVO-specific probes (NEB-HVO, dark grey); NEBNext with bacterial probes (NEB-B, grey); no removal (none, light grey).

**Table 1 biomolecules-12-00682-t001:** Strains used in this study.

Name and Wild Type Strain	Species Abbreviation	Genotype Used Here	Reference Genome
*Halobacterium salinarum* NRC-1 [24]	HBT	Δ*ura3*	GCF_000006805.1_ASM680v1
*Haloferax volcanii* DS2 [31]	HVO	Δ*pyrE*	GCF_000025685.1_ASM2568v1
*Haloferax mediterranei* ATCC33500 [25]	HFX	Δ*pyrE*	GCF_000306765.2_ASM30676v2
*Haloarcula hispainca* DF60 [25]	HAH	Δ*pyrF*	GCF_000223905.1_ASM22390v1

**Table 2 biomolecules-12-00682-t002:** All media recipes used for test organisms in this study.

Name	Species Abbreviation	Ingredients (per L)	Supplement	pH
CM (rich media)	HBT	250 g NaCl (Fisher Chemicals, Hampton, NH, USA); 20 g MgSO_4_.7H_2_O (Fisher Chemicals); C_6_H_5_Na_3_O_7_.2H_2_O (Fisher Chemicals); 2 g KCl (Fisher Chemicals); 10 g bacteriological peptone (Oxoid, Hampshire, UK)	50 mL uracil (1 mg/mL) (Acros Organics, Geel, Belgium)	6.8
YPC 18% (rich media [32])	HVO and HFX	144 g NaCl (Fisher Chemicals);4.2 g KCl (Fisher Chemicals); 18 g MgCl_2_.6 H_2_O (Fisher Chemicals); 20 g MgSO_4_.7H_2_O (Fisher Chemicals); 12 mL 1 M TrisHCl (Fisher Chemicals) pH7.5; 5 g yeast extract (Fisher Chemicals); 1 g 10 g bacteriological peptone (Oxoid); 1 g Cas aminoacids (VWR, Radnor, PA, USA)	50 mL uracil (1 mg/mL) (Acros Organics)	7.5
PR 18% (minimal media [this study])	HVO	170 g NaCl (Fisher Chemicals); 70 g MgCl_2_.6 H_2_O (Fisher Chemicals); 7 g KCl (Fisher Chemicals); 5 mL 1 M TrisHCl (Fisher Chemicals) pH7.5; 5 mL 1 M NH_4_Cl; 2 mL 0.25 M K_2_HPO_4_; 5 mL 1 M NaHCO3; 0.8 mL thiamine (1 mg/mL); 0.1 mL biotine (1 mg/mL); 0.5% glucose.	50 mL uracil (1 mg/mL) (Acros Organics)	7.2
YPC 23% (rich media [33])	HAH	180 g NaCl (Fisher Chemicals); 4.2 g KCl (Fisher Chemicals); 18 g MgCl_2_.6 H_2_O (Fisher Chemicals); 20 g MgSO_4_.7H_2_O (Fisher Chemicals); 12 mL 1 M TrisHCl (Fisher Chemicals) pH7.5; 5 g yeast extract (Fisher Chemicals); 1 g 10 g bacteriological peptone (Oxoid); 1 g Cas aminoacids (VWR)	50 mL uracil (1 mg/mL) (Acros Organics)	7.5

**Table 3 biomolecules-12-00682-t003:** Doubling time and incubation time for different species in different media.

Species	Media	Doubling Time (h)	Days Until Stationary Phase
HBT	CM	6	3
HVO	YPC18%	3	2.5 (36 h)
HVO	PR18%	12	3
HFX	YPC18%	2.5	2
HAH	YPC23%	6	3

**Table 4 biomolecules-12-00682-t004:** Primers used to check for genomic contamination.

Species	Forward Primer Sequence 5′-3′	Reverse Primer Sequence 5′-3′	Fragment Size
HBT	CGACATTCGGGTTGCGTTGTG	GGCGTTGTTCACGAAGCA	1372
HFX	CACATCAGCGAGGAGTTTGA	GACAGACGACGAGTTGGTCA	162
HVO	AGAAGTACAAGGGCGTCGAA	TTTTCGAACTCCTCGCTGAT	171
HAH	GCCGATTGCTCCGTCTACTA	ACTGCTCGGTGAGAAACGTC	161

**Table 5 biomolecules-12-00682-t005:** rRNA-coding gene identifiers for each species of interest.

Species	rRNA Type	Gene Identifier (s)	Alternate Gene Identifier (s)
HBT	16S	VNG_RS09790	VNG_r02
	23S	VNG_RS09800	VNG_r03
	5S	VHG_RS00395	VNG_r04
HVO	16S	HVO_RS13015, HVO_RS18920	HVO_3038, HVO_3064
	23S	HVO_RS13025, HVO_RS18910	HVO_3040, HVO_3062
	5S	HVO_RS13030, VHO_RS18905	HVO_3041, HVO_3061
HFX	16S	HFX_RS14380, HFX_RS08900	HFX_1820, HFX_2933
	23S	HFX_RS14370, HFX_RS08910	HFX_1822, HFX_2931
	5S	HFX_RS08915, HFX_RS14365	HFX_2930, HFX_1823
HAH	16S	HAH_RS08910, HAH_RS01110	HAH_1834, HAH_0232
	23S	HAH_RS08905, HAH_RS01120	HAH_1833, HAH_0234
	5S	HAH_RS08900, HAH_RS01125	HAH_1832, HAH_0235

## Data Availability

Code used to analyze data associated with this study is freely accessible via https://github.com/amyschmid/rRNA_analysis. RNA-seq data, including raw sequencing data and metadata, are available through the National Center for Biotechnology Information Gene Expression Omnibus (NCBI GEO) database at accession GSE200776.

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
