# Peer review of "Comparative Analysis of rRNA Removal Methods for RNA-Seq Differential Expression in Halophilic Archaea"

_biomolecules, 2022, doi:10.3390/biom12050682_

Round 1

Reviewer 1 Report

As stated by the authors, rRNA depletion is essential for transcriptomic analysis. The discontinuation of the Illumina RiboZero kit has left the archaeal community with several choices for ribo-depletion but there is a lack of data to evaluate these methods. The MS from Martinez-Pastor fills this gap, providing a comprehensive study of ribo-depletion methods for a selection of haloarchaea.

Overall the study is well documented, referenced, and with extensive sequencing data and strong statistical analysis. The conclusions are well supported by the data, the figures are appropriate, and the data corresponding to each figure is a welcome addition to Table S1.

While codes used to analyze the data and generate the figures are provided via links to GitHub, I could not find the accession numbers for the sequencing data. Sequence data and metadata should be released in a publicly accessible database and this information provided in the MS.

There is also quite a bit of confusion and discrepancies between supplementary tables and figures with two Table S3 and two Table S4 with data corresponding to the wrong figures; for example, what does Table S4-gDNA correspond to? Is Table S5 the same as Table S4?

Specific comments:

Please add total read counts to Table S1 to give a broader perspective to the users.

Why is the data for rP-PA not in Fig. 5?

Please specify if the Pan-archaea probe is Pan-archaea 27, as these probes evolve over time.

Update link for the power optimization tool Scotty.

L119-121: confusing, please rewrite

L235-239: if the intention was to continue ongoing studies with HBT, why use probes designed against HVO; please clarify.

L252-253: it would be useful here to add the total number of reads obtained for 12 multiplexed samples (or per sample), in addition to read numbers mapped to the genome and to non-rRNA genes.

L304-306: the rP-HVO method does not involve rRNA digestion, please rewrite.

L343: could the authors explain why the correlations of all methods to RZ are lower than between other methods?

L372: change to “Discussion and Conclusions”

Reviewer 2 Report

Martinez-Pastor et al have carried out an evaluation of different methods for rRNA removal prior to RNA-seq, using different halophilic archaeal model species. The manuscript is very thorough and comprehensive, and the conclusions will be of great utility for researchers in the field. It is particularly commendable that the authors have used four different model species of haloarchaea, thereby increasing the value of their results.

The methods and results could be improved as follows:

  • Main comment:
    • Given that the principal conclusion of the study is that commercial rRNA depletion kits work efficiently, so long as species-specific probes are used, it would be of great utility to make available to researchers a list of the recommended probes for each of the four species examined. I suggest that this list of recommended probes be provided in the main manuscript.
  • Minor comments:
    • The tab in Table S2 is titled Table S5.
    • Table S3 is duplicated, please remove the redundant copy.
    • Table S4 is present twice but the files are very different, TableS4_gDNA does not appear to be the correct file because one of the tabs is titles Table S3, should this file and/or tab have a different name?
    • The title for Figure S1 at the top of the page could be tidied up.
    • Would it be appropriate to comment on costs of each kit?
